# Nutritional Trade-Offs in *Drosophila melanogaster*

**DOI:** 10.3390/biology14040384

**Published:** 2025-04-07

**Authors:** Juliano Morimoto

**Affiliations:** 1Institute of Mathematics, School of Natural and Computing Sciences, University of Aberdeen, Fraser Noble Building, Aberdeen AB24 3UE, UK; juliano.morimoto@abdn.ac.uk; 2Programa de Pós-graduação em Ecologia e Conservação, Universidade Federal do Paraná, Curitiba 82590-300, Brazil

**Keywords:** comparative nutrition, dipterans, energy metabolism, gene expression

## Abstract

Animals regulate their nutrient intake to meet their physiological needs, but no single diet optimizes all traits, creating the potential for nutritional trade-offs. This study analyzed how different protein-to-carbohydrate (PC) ratios affect traits in the fruit fly *Drosophila melanogaster* using data from the Geometric Framework of Nutrition. Three nutrient regions were identified: low PC ratios (e.g., 1:8), which support longer lifespans but hinder growth and reproduction; high PC ratios (e.g., 1:1), which enhance development, body mass, and male reproduction but reduce lifespan; and intermediate PC ratios (<1:1 to >1:8), which maximize female reproduction and larval survival. These findings highlight trade-offs between lifespan and reproduction, suggest metamorphosis may help balance nutrient needs across life stages, and point to potential genetic conflicts between males and females over metabolic traits. This research advances our understanding of how animals respond to their diets to optimize specific traits, addressing key questions in evolutionary biology and health.

## 1. Introduction

Animals acquire nutrients to subsidize their metabolic demands, but the quantity and ratio of these nutrients vary [1,2,3,4]. When animals cannot acquire nutrients in the quantity and ratio for all traits, there is the potential for conflict whereby animals need to balance their nutrient intake to maximize one trait at the expense of another (i.e., “nutritional trade-offs”) [5,6,7]. Previous research has uncovered evidence to support the concept of nutritional trade-offs between reproduction and lifespan and reproduction and immune traits [5,6,7,8,9,10,11,12,13] and between reproductive traits that contribute to different sexual selection episodes [14,15,16] (see also [17]).

Nutritional trade-offs are likely ubiquitous and could play an important role in shaping animal nutrition and evolution. However, previous work has largely studied nutritional trade-offs among only a limited number of traits (usually two or three), with few exceptions (see e.g., [18]). For example, in a landmark paper, Lee et al. [6] used the Geometric Framework of Nutrition (GF) to comprehensively assess how the ratio of proteins and carbohydrates (PC ratio) in diets modulated lifespan, reproductive rate, and lifetime egg production in female *Drosophila melanogaster* Meigen (1830). Likewise, Maklakov et al. [7] studied PC ratio effects on the expression of three traits (lifespan and two reproductive traits) in each sex of the cricket *Teleogryllus commodus* Walker (1869). These examples are representative of the wider literature (e.g., [10,12,14,15,16,19,20,21] and references therein) and continue to stimulate new studies that uncover insights into animal nutritional ecology in both basic and applied sciences [22,23,24,25]. Yet, it is also important to step back to try and unify our knowledge in a more general context to gain a proper overview of nutritional trade-offs across multiple traits within and between species [10,26]. In this regard, there are unmatched advantages to working with model organisms like *Drosophila melanogaster* for which the nutritional effects have been mapped across several traits in high resolution using the GF. By compiling what we know about *D. melanogaster* responses to diets from GF studies—which to my knowledge has never been done—we will gain the much-needed general insight into nutritional trade-offs that will help us interpret current knowledge and guide future work.

Therefore, in this study, I collated the PC ratio of diets that maximize (best) or minimize (worst) a wide range of traits in *D. melanogaster*, using my previous analytical methods to reconstruct and analyze GF landscapes [27,28]. My main goal was to characterize the potential for nutritional trade-offs by highlighting traits that have opposing responses to the same PC ratios. To achieve this, my assumption was that the *Drosophila* strains used across different studies responded were comparable. This assumption was necessary because I do not have the information of the genetic architecture of all lines in the published literature. It follows from this assumption that *Drosophila* strains respond similarly to different diets irrespective of genetic differences, which we know is not always true at least for highly inbred lines (e.g., [24]). I discuss the implications of this in the methods and discussion sections. Nevertheless, the findings of this work highlight the potential for nutritional trade-offs in *Drosophila* that will stimulate future work to uncover the causes and consequences for the ecology and evolution of this and other species.

## 2. Materials and Methods

### 2.1. The Data

I included studies from the literature which used *Drosophila melanogaster* as a model system and the Geometric Framework for Nutrition as the experimental design. I included studies that measured both food intake and that manipulated the macronutrients in the diets without measuring diet intake; the former approach was usually adopted by studies using adult flies while for the latter this was larvae. I also included studies with liquid diets that used the CAFÉ assay and studies with solid media (Table 1). When possible, I used studies for which raw data was made available in the original publication. When the raw data was not available for lifespan, I used my previously validated approach to reconstruct GF landscapes to extract data that could be used to estimate PC ratios [27]. Briefly, each GF landscape is segmented from an image obtained in the original publication and then reconstructed with a semi-automated algorithm capable of building a surrogate GF landscape with peaks and valleys in the same region as the original landscape. This allows us to analyze GF landscapes for which the raw data is unavailable [27]. Table 1 lists the studies which were used and the traits that were studied. Raw data are provided in Appendix A in Appendix A.

### 2.2. Estimates of the Peaks and Valleys

Using R version 4.3.2 [33], I used the Nutrigonometry models to estimate the best (peaks) and worst (valleys) diets for the expression of the traits [28]. I plotted the average protein and carbohydrate estimate of peaks and valleys from different traits in the same nutrient space to aid visualization of the potential nutritional trade-offs among traits. I estimated peaks and valleys for all the studies individually and then averaged these estimates across different studies that measured the same trait to create a single estimate of PC ratios for the peak and valley (e.g., male lifespan in [19,30]). The best and worst PC ratio for male paternity share was estimated from [16] when males were the first (Paternity 1) or second (Paternity 2) to mate with females. I also estimated ‘refractoriness’ as the latency of females mated to focal males to remate with a competitor male when focal males were the first to mate (i.e., which helps increase male paternity 1) and the latency of previously mated females to remate with a focal male (‘Latency (Remate)’) when focal males were the second to mate (i.e., for males to gain paternity 2) [see [16] for details]. All estimates of peaks and valleys across traits were plotted in milligrams. There were no outliers in the data. All figures were done using the ‘ggplot2’ package version 3.5.1 [34]. As mentioned above, studies varied in multiple ways, namely the genetic background of the *Drosophila melanogaster* stock, diet composition (solid vs. liquid), and intake estimates. I therefore opted to not conduct statistical inferences as those would inevitably be biased. Note that although traditional GF experiments enable caloric intake and macronutrient ratio effects to be disentangled, this is not possible in this cross-study comparison due to confounding factors. Nevertheless, even if caloric intake had an influence, it would likely affect the position of the points relative to the origin along an isoline and not change the ratio of nutrients that maximize (or minimize) a particular trait [3,4].

## 3. Results

### 3.1. The Distribution of Peaks and Valleys in the Nutrient Space Created Three Regions

#### 3.1.1. Region 1: High Carbohydrate, Low Protein Diets

Low PC ratios (PC ratio of ~1:8 or lower) maximized lifespans for both adult males and females as well as short-term female oviposition rates and male refractoriness (Figure 1). On the other hand, low PC ratios minimized male paternity 1, male and female adult body mass, larval developmental time and survival, and female ovariole number. Overall, low PC ratios maximize lifespans at the expense of most traits related to growth and male and female reproduction.

#### 3.1.2. Region 2: High Protein, Low Carbohydrate Diets

High PC ratios (PC ratios of ~1:1 or higher) maximized larval developmental time, the adult body masses of both sexes, female ovariole numbers and, for diets with a higher concentration of macronutrients, male paternity 2 and the latency of females to remate with focal males (Figure 1). On the other hand, high PC ratios minimized lifespans in both sexes as well as short-term female oviposition rates. Male paternity 2 and the latency of females to remate with focal males were also minimized in this region, but when diets had less macronutrients. Overall, high PC ratios maximized traits related to growth and reproduction at the expense of lifespan.

#### 3.1.3. Region 3: Balanced Diets

Balanced diets with PC ratios between ~1:1 and ~1:3 maximized larval survival and female reproductive rates and lifetime egg production. No trait was minimized in this region (Figure 1).

## 4. Discussion

Animals must balance their nutrient intake to express fitness-related traits, which creates the potential for nutritional trade-offs among traits with competing nutritional needs [35]. Using the Nutrigonometry models on key GF studies from the *Drosophila melanogaster* literature, I compiled information about the optimum PC ratio for a wide range of traits across life stages of the fly *D. melanogaster* and found a strong potential for nutritional trade-offs among traits related to lifespans, growth and reproduction. Specifically, there were three regions in protein–carbohydrate nutrient space where peaks and valleys of traits were found. Low PC ratios, which are diets richer in carbohydrates, maximized lifespans and short-term female oviposition rates but minimized all traits related to larval growth and survival, adult body mass, and adult reproduction. High PC ratios, which are diets richer in protein, showed the opposite effect. Three traits were maximized at more intermediate PC ratios, namely female lifetime egg production (PC ~1:3), female reproductive rates (PC~1:2), and larval survival (PC ~1:1.5) (Figure 1). Flies are holometabolous insects and metamorphosis might help resolve nutritional trade-offs between life stages [36,37]. This is less clear within life stages and between sexes, as shown here for male and female reproductive and lifespan traits. Because males and females share the same genome, nutritional trade-offs could create the potential for intralocus sexual conflict [38,39,40], which might be pervasive across insects [27]. One way that intralocus sexual conflict could be resolved is through the modulation of the expression of metabolic genes in males and females [41], but we do not yet have a complete understanding of how nutritional trade-offs and sexual conflict interact to modulate organism-wide gene expression (but see discussion below).

It would be interesting to study the molecular mechanisms and metabolic pathways which are up- and down-regulated when flies experience different diets. An ambitious but worthwhile goal is to create GF performance landscapes of genes and pathways using omics technologies to give us the necessary mechanistic insights into the drivers underlying nutritional trade-offs. The TOR and AMPK pathways are two obvious higher level regulatory pathways that control nutritional trade-offs but what are the genes that modulate nutritional responses and trade-offs downstream of these major pathways [42,43,44,45]? My previous work has raised the possibility that the uric acid pathway modulates at least some diet- and density-dependent responses during *D. melanogaster* larval development [46], but that study lacks the high-resolution nature of GF experiments and is by no means comprehensive. Other studies in insects have used the GF but did not gain similar levels of molecular insights in either larvae (e.g., [18,20,21,32]) or adults (e.g., [6,19,30]). Molecular insights are crucial because we are now uncovering how diet composition interacts with genes and their expression to modulate diet responses, growth, and fitness. For example, Yurkevych et al., [47] have shown changes in the expression of a wider range of genes that underpinned tolerance to high protein diets. Similarly, Francis et al. [23] have shown that genetics plays a major role in diet-dependent responses in *D. melanogaster*. Likewise, Havula et al. [24] have shown that genetics can strongly modulate larval development and survival, particularly in less favorable diets such as high-sugar diets. Investigating the effects of larval crowding—which is known to modulate protein availability—on *D. melanogaster* larval gene expression, I found transcriptomic-wide trade-offs across most major pathways including metabolism and immunity [48]. Similar findings were reported in *Drosophila simulans* [49]. It is possible that these transcriptomic-wide trade-offs emerge and are modulated by diets, but specifically how remains to be ascertained. Future work should take advantage of the molecular resources available for *D. melanogaster* to uncover further insights into gene–diet interactions.

The data collated here highlight a nutritional conundrum for adult flies: how to balance diets that maximize lifespans against those that enhance reproduction. Physiological nutrient requirements vary depending on life stage, environment, genetic background, and stress levels, making nutritional choices inherently multi-factorial [35]. However, *Drosophila melanogaster* possess a well-characterized molecular pathway that acts as a dietary switch: the sex peptide (SP) pathway [50]. When females mate, they receive a cocktail of seminal fluid proteins from the male’s ejaculate, the most abundant of which is SP [51]. SP binds to its receptor (sex peptide receptor or SPR) in the female reproductive tract and brain, triggering physiological and behavioral changes, including increased protein intake [52]. Thus, mating serves as a clear biological signal that reproduction should be prioritized at least partly through the physiological cascade activated by the SP–SPR interaction. Males also express SPR in the brain, but whether this influences their nutritional needs and behavior remains unclear. *Drosophila melanogaster* males are reproductively active even in the absence of suitable mates, which suggests that they may be inherently predisposed to prioritizing diets that support reproduction over longevity [53,54].

Based on this, I hypothesize that males and females have different baseline nutritional needs: unmated females require a lower PC ratio until mating triggers a shift towards higher PC ratio diets that enhance reproductive output. In contrast, males may be hardwired to consume higher PC ratio diets by default, unless environmental factors (e.g., low female encounter rates) drive a shift towards diets that favor lifespan extension (Figure 2a). This hypothesis has significant implications for sexual conflict over nutrition. In a *D. melanogaster* population that adheres to Bateman’s principles—which appears ubiquitous—most females are mated, with fitness constrained only by egg production, whereas a smaller proportion of males successfully mate, leaving many unmated [55]. Under these conditions, sexual conflict over nutrition is likely weak because the optimal PC ratio for both sexes tend to converge: mated females shift to higher PC ratios to maximize reproduction, while males already prioritize these diets for reproductive success (Figure 2b). Further studies are needed to test this hypothesis and, more broadly, to investigate how population structures, mating systems, and evolutionary pressures, including Bateman’s principles, shape nutritional responses and adaptation.

Our data reflected studies that used a wide range of methodological details, such food types (solid vs. liquid), which might influence nutritional responses. For example, measuring diet intake and compensatory responses when animals are given a solid diet remains challenging. The CAFÉ assay enables a precise quantification of dietary intakes; however, it provides a highly unnatural way of feeding for the flies, which could lead to other physiological and behavioral changes that affect their response to diets. We still lack a systematic comparison of the nutritional responses of flies with similar genetic backgrounds on different diet types or different genetic backgrounds on the same diet type to properly quantify the impact of this experimental approach on nutritional responses. With a growing number of studies in the literature, it will become possible to quantify these effects and account for their consequences, leading to better interpretation of results. Nevertheless, it is worth noting that GF landscapes for lifespans and reproduction are qualitatively similar among different studies (Appendix A), which partly assuages concerns about the confounding effects of unmeasured factors on the nutritional patterns presented here.

Even though there is growing evidence that genetics play an important role in diet-dependent responses, I assumed that the nutritional responses of the different *D. melanogaster* strains used across studies in the literature are comparable. There are not enough studies which consistently use the same genetic strain to allow for a study such as this. Therefore, the results presented here should be interpreted with caution, as there are likely unaccounted for genetic effects underpinning the estimates for peaks and valleys. However, it is interesting that higher larval survival variability was reported in high-sugar diets across 196 *D. melanogaster* isolines [24]. As shown here, this could be explained by the fact that diets with low PC ratios (i.e., sugar-rich diets) represent the worst diet for larval survival and therefore impose a much harsher developmental environment that could translate into higher variability in survival. Similar effects of genetic variability in more extreme (high sugar and high protein) diets have also been reported in other studies [23]. Together, these findings agree with the argument I recently put forward that trait variability should increase when organisms feed in imbalanced diets [56]. This remains to be empirically tested.

## 5. Conclusions

Through data collection from a range of studies, I demonstrated that larval and adult traits in *D. melanogaster* have distinct—often conflicting—nutritional requirements, creating the potential for nutritional trade-offs and sexual conflict over nutrition. Specifically, PC ratios that maximized lifespans almost invariably minimized reproductive traits, particularly in females. Future studies should focus on uncovering the mechanisms that trigger and enable dietary switches that optimize different life-history strategies. Moreover, a broader taxonomic perspective is needed to understand how dietary responses are realized and whether phylogenetic patterns across taxa can provide deeper insights into the evolutionary dynamics of nutritional adaptation.

## Figures and Tables

**Figure 1 biology-14-00384-f001:**
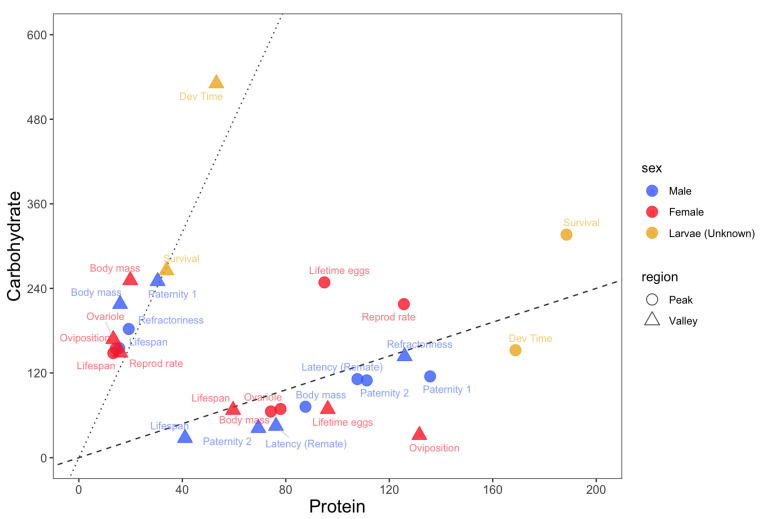
**Nutritional trade-offs in *Drosophila melanogaster***. The protein (*x*-axis) and carbohydrate (*y*-axis) estimates for the peaks (circles) and valleys (triangles) of each trait. Light blue = adult males; Red = adult females; Green = larvae. Black dashed line represents a PC ratio of 1:1.2. Black dotted line represents a PC ratio of 1:8.

**Figure 2 biology-14-00384-f002:**
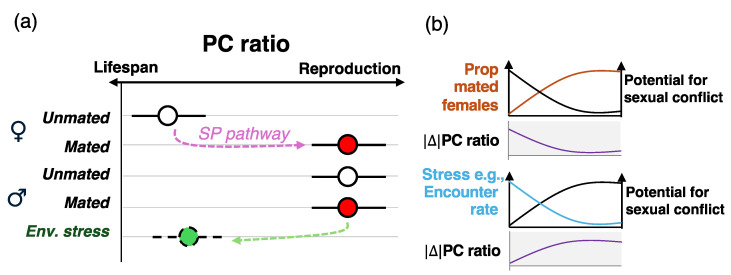
**Nutritional switches might assuage the potential for nutritional trade-offs.** (**a**) Upon mating, females shift their dietary preference and consume higher PC ratio diets to support egg production, as a result of the sex peptide (SP) pathway. Males eat higher PC ratios regardless of their mating status, maximizing reproductive traits as opposed to lifespans. I hypothesize that this is the default nutritional behavior in females and males, respectively, and males only shift to PC ratios that maximize lifespans due to environmental conditions, which could lead to males surviving longer and achieving higher fitness. An example could be a population where female encounter rates by males are relatively low, and males could benefit from living longer to maximize their chances of reproduction. (**b**) Example of two population factors that can modulate the potential for sexual conflict over nutrition. As the proportion of mated females increases, the absolute difference |Δ| of the PC ratio between males and females decreases (i.e., both eat higher PC ratio diets), in turn leading to lower potential for sexual conflict. Stress, such as for example low encouter rates, could lead males to shift their dietary intake to lower PC ratio diets to live longer and increase their chances of findings a mate, increasing |Δ| and the potential for sexual conflict. This hypothesis remains untested.

**Table 1 biology-14-00384-t001:** Literature that was included in this study.

Lead Author	Year	Trait(s)	Stage	Sex	Strain	Diet	Reference
Lee	2008	Lifespan, reproductive rate, lifetime egg production	Adults	Females	Canton-S	Liquid and yeast-based	[6]
Semaniuk	2018	Lifespan	Adults	Females	‘IF’		[29]
Jensen	2015	Lifespan	Adults	Females, Males	Dahomey (Stuart Wigby)	Liquid and chemically-defined	[19]
Carey	2022	Lifespan	Adults	Females, Males	Dahomey (Nick Priest)	Solid and chemically-defined	[30]
Rodrigues	2015	Adult body mass, Ovariole number	Adults	Females, Males	Outbred (Azeitão, Portugal)	Solid and yeast-based	[18]
Lihoreau	2016	Oviposition	Adults	Females	Canton-S	Solid and yeast-based	[31]
Morimoto	2016	Paternity share (P1 and P2), Refractoriness *, Latency to remate	Adults	Males	Dahomey	Liquid and yeast-based	[16]
Kutz	2019	Developmental time, survival	Larvae	NA	Outbred (Ballina, Australia)	Solid and yeast-based	[32]

* There was no detectable statistical effect of diet for this trait in the original study by [16] but the trait was included here for completeness.

## Data Availability

The data used in this paper is available in the Appendix A.

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
