# Peer review of "Nutritional Trade-Offs in Drosophila melanogaster"

_biology, 2025, doi:10.3390/biology14040384_

Round 1
Reviewer 1 Report
Comments and Suggestions for Authors
This manuscript examines the effects of different protein:carbohydrate ratios on various traits in fruit flies by summarising studies using the geometric framework of nutrition. The aim of the communication is to elucidate how these dietary trade-offs affect lifespan, reproduction and other traits.
In my opinion, the conclusion drawn by the author is not supported by the data. Different fly strains may have inherent differences in their nutritional intake, requirements and responses, making comparison between them unreliable. Assessment of fly dietary intake is essential in this type of study. As some of the studies included in this communication lack this data, it is difficult to determine whether the observed effects on lifespan are due to caloric restriction rather than specific dietary ratios. Caloric restriction is well known to extend lifespan in many organisms, and without controlling for this variable, conclusions about protein:carbohydrate ratios may be misleading. In addition, no distinction was made between the protein sources used (yeast vs. amino acids). Different dietary components may have different metabolic effects. For example, yeast provides a complex mixture of nutrients beyond just protein, whereas single amino acid diets provide a more controlled but potentially less naturalistic environment. The consistency of the diet also makes a difference. Liquid versus solid diets can affect the amount and rate of food intake, which in turn affects nutrient absorption and metabolic responses. This variability adds another layer of complexity that is not accounted for in the analysis.
Although the manuscript addresses an interesting topic in nutritional ecology, the compilation of several studies conducted with different and incomparable methodological and analytical approaches does not provide robust results and therefore cannot be considered for publication.
Author Response
Reviewer 1
This manuscript examines the effects of different protein:carbohydrate ratios on various traits in fruit flies by summarising studies using the geometric framework of nutrition. The aim of the communication is to elucidate how these dietary trade-offs affect lifespan, reproduction and other traits.
Reply: Thank you. I just want to correct the reviewer. The aim is not to “elucidate how these dietary trade-offs affect lifespan, reproduction, and other traits”. The aim of the manuscript was to “characterize the potential for nutritional trade-offs by highlighting traits that have opposing responses to the same PC ratios (Lines 69-71 in the original submission).
This distinction is crucial for the proper contextualization of the results presented because I am not attempting to provide functional, evolutionary, or mechanistic understanding of the consequences of the nutritional trade-offs.
Instead, my aim is to provide the first overview across multiple studies of the existence (or not) of nutritional trade-offs across multiple traits. Contrary to what the reviewer suggests, I do not present any analysis that claims to “elucidate how these trade-offs affect lifespan, reproduction and other traits”. Much of the reviewers’ criticism impinges on this misunderstanding.
In my opinion, the conclusion drawn by the author is not supported by the data. Different fly strains may have inherent differences in their nutritional intake, requirements and responses, making comparison between them unreliable.
Reply: I disagree. It is true that different studies have different methods, strains etc that make direct comparisons unreliable (hence why I originally stated that statistical inference was not conducted here). However, the effect sizes can be compared. In this work, I present a qualitative assessment of the literature using the Geometric Framework in D. melanogaster life-histories, and such qualitative comparison is – and should be, if science is reproducible – reliable.
To appease the reviewer, I have analysed the variability on the estimates of lifespan and reproduction among four studies conducted with different genetic lines and diet texture and recipe. As shown below, there is no qualitative changes in the results: lifespan is maximized at a much lower PC ratio than reproduction. Similar comparison cannot be done for all traits because data is not available, but this stresses that as far as the data exist, the qualitative comparisons presented here are robust and consistent.

It is important to highlight that this limitation was clearly and repeatedly highlighted in the original submission:
Lines 69-77 (Introduction)
To achieve this, my assumption was that the Drosophila strains used across different studies responded were comparable. This assumption was necessary because I do not have the information of the genetic architecture of all lines in the published literature. It follows from this assumption that Drosophila strains respond similarly to different diets irrespective of genetic differences, which we know is not always true at least for highly inbred lines (e.g., [24]). I discuss the implications of this in the methods and discussion sections.
Lines 114-116 (Methods)
As mentioned above, studies varied in multiple ways: genetic background of the Drosophila stock, diet composition (solid vs liquid), intake estimates. I therefore opted to not conduct statistical inferences as those would inevitably be biased.
Lines 193-206 (Discussion)
Even though there is growing evidence that genetics play an important role in diet-dependent responses, I assumed that the nutritional responses of the different D. melanogaster strains used across studies in the literature are comparable. There are not enough studies which consistently use the same genetic strain to allow for a study such as this. Therefore, the results presented here should be interpreted with caution, as there are likely unaccounted genetic effects underpinning the estimates of peaks and valleys. However, it is interesting that Havula et al., [24] reported higher larval survival variability in high-sugar diets across 196 D. melanogaster isolines. As shown here, this could be explained by the fact that diets with low PC ratio (i.e., sugar-rich diets) represent the worst diet for larval survival and therefore impose a much harsher developmental environment that could translate into higher variability in survival. Francis et al., [23] also found similar effect of genetic variability in more extreme (high sugar and high protein) diets. Together, these findings agree the argument I recently put forward that trait variability should increase when organisms feed in imbalanced diets [50]. This remains to be empirically tested.
Assessment of fly dietary intake is essential in this type of study. As some of the studies included in this communication lack this data, it is difficult to determine whether the observed effects on lifespan are due to caloric restriction rather than specific dietary ratios. Caloric restriction is well known to extend lifespan in many organisms, and without controlling for this variable, conclusions about protein:carbohydrate ratios may be misleading.
Reply: Contrary to the reviewers’ comment, dietary intake is not essential in GF experiments. The GF enables us to parse the effects of calories vs macronutrient ratio (Simpson and Raubenheimer, 1993; Raubenheimer and Simpson, 1993). This is a well-established advantage of this framework (e.g., Simpson and Raubenheimer, 2012).
In the present study, I refrained from making inferences about caloric effects because different studies had different performance landscape resolutions (sensu Morimoto, 2022). Thus, my data does not allow for comparisons that involve caloric intake, but it does allow us to compare the macronutrient ratio in which traits are maximized or minimized. I have now made this even more explicit in the methods section (Lines 117 to 121):
“Note that although traditional GF experiments enable caloric intake and macronutrient ratio effects to be disentangled, this is not possible in this cross-study comparison due to confounding factors. Nevertheless, even if caloric intake had an influence, it would likely affect the position of the points relative to the origin along an isoline and not change the ratio of nutrients that maximise (or minimize) a particular trait [3,4]”.
In addition, no distinction was made between the protein sources used (yeast vs. amino acids). Different dietary components may have different metabolic effects. For example, yeast provides a complex mixture of nutrients beyond just protein, whereas single amino acid diets provide a more controlled but potentially less naturalistic environment.
Reply: This information was now added to Table 1.
The consistency of the diet also makes a difference. Liquid versus solid diets can affect the amount and rate of food intake, which in turn affects nutrient absorption and metabolic responses. This variability adds another layer of complexity that is not accounted for in the analysis.
Reply: This information was now added to Table 1. As explicitly mentioned in the Methods, there were no statistical inferences precisely for these reasons. Instead, this short communication is a qualitative assessment of the current literature which assumes, without comprehensive evidence such as the one provided here, that nutritional trade-offs between traits exist.
Lines 114-116 (Methods)
“As mentioned above, studies varied in multiple ways: genetic background of the Drosophila stock, diet composition (solid vs liquid), intake estimates. I therefore opted to not conduct statistical inferences as those would inevitably be biased.”
Although the manuscript addresses an interesting topic in nutritional ecology, the compilation of several studies conducted with different and incomparable methodological and analytical approaches does not provide robust results and therefore cannot be considered for publication.
Reply: Thank you for reviewing the manuscript. It is difficult to ascertain what the reviewer means by “robust” since there were no statistical tests. Moreover, as a qualitative study, the results are consistent, and even the responses across strains and diet types show evidence of consistent results. Based on this, I would argue that the study is as robust as it was designed to be based on a qualitative assessment of the literature, compiling effect sizes across different studies.
Reviewer 2 Report
Comments and Suggestions for Authors
The authors studied how different protein-to-carbohydrate (PC) ratios affect traits in the fruit fly Drosophila melanogaster using data from the Geometric Framework of nutrition. The work is based on secondary data and is very brief. The work is moderately novel. There are some errors, and also my suggestions are mentioned below:
Line 11-12: Drosophila melanogaster---- will be in italicised form.
Line 15-16: intermediate PC ratios (1:1 to 1:8) ---- check the ratio. It will be (<1:1 & >1:8).
Line 38: PC ratios (between 1:1 and 1:8) ----- check.
Line 36: Keywords---- arrange alphabetically.
Line 46-47:Thus, there is a general assumption that nutritional trade-offs are ubiquitous----- lack of continuity. Rewrite clearly.
Line 51: the landmark paper by Lee et. al. [6] ----- rewrite.
Line 53-54: Drosophila melanogaster---- add author citation, like Drosophila melanogaster Meigen
Line 54: Maklakov et. al.---- Maklakov et al.
Line 209-215: Avoid references. Focus on key findings and the significance of the work.
Author Response
Reviewer 2
The authors studied how different protein-to-carbohydrate (PC) ratios affect traits in the fruit fly Drosophila melanogaster using data from the Geometric Framework of nutrition. The work is based on secondary data and is very brief. The work is moderately novel. There are some errors, and also my suggestions are mentioned below:
Reply: Thank you for comments that helped improve the work. As a “communications” paper, it was intended to be brief.
Line 11-12: Drosophila melanogaster---- will be in italicised form.
Reply: Done.
Line 15-16: intermediate PC ratios (1:1 to 1:8) ---- check the ratio. It will be (<1:1 & >1:8).
Reply: Done.
Line 38: PC ratios (between 1:1 and 1:8) ----- check.
Reply: Done.
Line 36: Keywords---- arrange alphabetically.
Reply: Done.
Line 46-47:Thus, there is a general assumption that nutritional trade-offs are ubiquitous----- lack of continuity. Rewrite clearly.
Reply: Done. I have now restructured the first two paragraphs for clarity. The beginning of the second paragraph now reads (Lines 48-51):
“Nutritional trade-offs are likely ubiquitous and could play an important role in shaping animal nutrition and evolution. However, previous work has largely studied nutritional trade-offs among only a limited number of traits (usually two or three), with few exceptions (see e.g., [18]).”
Line 51: the landmark paper by Lee et. al. [6] ----- rewrite.
Reply: Done. It reads (Lines 51-54):
“For example, in a landmark paper, Lee et. al. [6] used the Geometric Framework of nutrition (GF) to comprehensively assess how the ratio of protein and carbohydrate (PC ratio) in diets modulated lifespan, reproductive rate, and lifetime egg production in female Drosophila melanogaster.”
Line 53-54: Drosophila melanogaster---- add author citation, like Drosophila melanogaster Meigen
Reply: Done (Line 54)
Line 54: Maklakov et. al.---- Maklakov et al.
Reply: Corrected.
Line 209-215: Avoid references. Focus on key findings and the significance of the work.
Reply: Done. It now reads (lines 204-205 & 209 to 210):
“However, it is interesting that higher larval survival variability was reported in high-sugar diets across 196 D. melanogaster isolines [24]. As shown here, this could be explained by the fact that diets with low PC ratio (i.e., sugar-rich diets) represent the worst diet for larval survival and therefore impose a much harsher developmental environment that could translate into higher variability in survival. Similar effect of genetic variability in more extreme (high sugar and high protein) diets have also been reported in other studies [23]. Together, these findings agree the argument I recently put forward that trait variability should increase when organisms feed in imbalanced diets [50]. This remains to be empirically tested.”
Reviewer 3 Report
Comments and Suggestions for Authors
Line 83 - I would add references for these studies. Based on experience, it is good to give the reader and idea of how the other data was analyzed and how this could have impacted your analysis.
Line 100- Can you clarify if you removed outliers of the average you performed twice? What were these outliers? Performing averages may be masking the data
Discussion - I would add a table of genes related to cafe' assays data used in this paper. I would also do a pathway analysis like figure to show the molecular mechanisms if there is some strong correlation with the cafe assay. Is the transcriptomics data you mentioned performed in sync with a cafe assay experiment?
Comments on the Quality of English LanguageGood
Author Response
Line 83 - I would add references for these studies. Based on experience, it is good to give the reader and idea of how the other data was analyzed and how this could have impacted your analysis.
Reply: The references are given in Table 1 in the main text. It is not entirely clear what the reviewer means by “Based on experience, it is good to give the reader and idea of how the other data was analyzed and how this could have impacted your analysis”; specifically, I do not know that “the other data” is. I assumed that the reviewer means that a discussion on the possible caveats of the compiled data is warranted. This was given in the introduction (Lines 72-78), methods (Lines 115 – 118) and discussion (Lines 199 – 213) in the original submission.
Line 100- Can you clarify if you removed outliers of the average you performed twice? What were these outliers? Performing averages may be masking the data
Reply: There were no outliers in the data and I now made this explicit in Line 118.
Discussion - I would add a table of genes related to cafe' assays data used in this paper. I would also do a pathway analysis like figure to show the molecular mechanisms if there is some strong correlation with the cafe assay. Is the transcriptomics data you mentioned performed in sync with a cafe assay experiment?
Reply: It is not clear to me what the reviewer means by “add a table of genes related to cafe' assays data used in this paper”. This paper presents the nutritional analysis of life-history traits, not genes. Line 194 discussed my recent paper that used RNA-seq to quantify trade-offs at the gene expression level in larvae, for which café assay is impossible (i.e., larvae cannot feed on capillaries). I highlighted this paper along with that of others (e.g., Havula and Francis) in the discussion to make a point about the need and gains of molecular studies. I cannot do the “pathway analysis like figure” because the paper does not present gene expression data, nor the studies that conducted RNAseq were done using the café assay.
Reviewer 4 Report
Comments and Suggestions for Authors
Despite the manuscript's clear title, a significant discrepancy exists between the title and its actual content. The author's reference to the genus Drosophila in the title could mislead readers about the study's scope.
While the genus Drosophila is diverse, with over 1,500 species exhibiting a wide range of morphological, behavioral, and breeding characteristics, the author's study focuses on a single species. This deliberate focus does not allow for a deep and thorough exploration of this species despite the study being based on information from only eight published articles and using Nutrigonometry models to estimate the best and worst diets for trait expression.
Conducting research based on the results of other studies, which often use different methods and procedures, can be complicated.
What is also striking is the use of only eight articles, given that Drosophila melanogaster is one of the most studied insects in the world. This limited number of sources raises concerns about potential biases and the comprehensiveness of the study's findings.
The results of the research are summarized in a single graph
Despite the manuscript's clear title, a significant discrepancy exists between the title and its actual content. The author's reference to the genus Drosophila in the title could mislead readers about the study's scope.
While the genus Drosophila is diverse, with over 1,500 species exhibiting a wide range of morphological, behavioral, and breeding characteristics, the author's study focuses on a single species. This deliberate focus does not allow for a deep and thorough exploration of this species despite the study being based on information from only eight published articles and using Nutrigonometry models to estimate the best and worst diets for trait expression.
Conducting research based on the results of other studies, which often use different methods and procedures, can be complicated.
What is also striking is the use of only eight articles, given that Drosophila melanogaster is one of the most studied insects in the world. This limited number of sources raises concerns about potential biases and the comprehensiveness of the study's findings.
The results of the research are summarized in a single graph
Author Response
Reviewer 3
Despite the manuscript's clear title, a significant discrepancy exists between the title and its actual content. The author's reference to the genus Drosophila in the title could mislead readers about the study's scope.
Reply: I have now updated the title to: “Nutritional trade-offs in Drosophila melanogaster” to reflect the species target of this study.
While the genus Drosophila is diverse, with over 1,500 species exhibiting a wide range of morphological, behavioral, and breeding characteristics, the author's study focuses on a single species. This deliberate focus does not allow for a deep and thorough exploration of this species despite the study being based on information from only eight published articles and using Nutrigonometry models to estimate the best and worst diets for trait expression.
Conducting research based on the results of other studies, which often use different methods and procedures, can be complicated.
Reply: Thanks for the overview of the Drosophila genus. I have now changed the title to include “Drosophila melanogaster” which better reflect the scope of this work.
What is also striking is the use of only eight articles, given that Drosophila melanogaster is one of the most studied insects in the world. This limited number of sources raises concerns about potential biases and the comprehensiveness of the study's findings.
Reply: It is important to clarify one key point which was mentioned in the original manuscript (Lines 61 – 66 and Lines 81-83 in the original submission): only studies which used the geometric framework were included in this study. This narrow scope was designed because only the Geometric Framework enables one to disentangle the effects of macronutrients and calories, and to experiment on multiple nutrients simultaneously.
There are only seven fly species for which the Geometric Framework has been used in its full design to measure life history traits related to lifespan and reproduction and development (D. melanogaster, D. simulans, Zaprionus indianus, Bactrocera tryoni, Ceratitis cosyra, Anastrepha ludens, Hermetia illuscens). For D. simulans and Z. indianus only developmental data (i.e. no lifespan or reproduction) is available in the GF design.
All studies that used the Geometric Framework to measure Drosophila melanogaster lifespan and reproduction were added to our study making its scope, albeit narrow, comprehensive. For the development, I omitted the work by Ma et al. (2022) because their experimental design involved manipulations of amino acids to match the fly exome. I also omitted the work by Zanco et al., (2023), where I am a co-author, because is not currently peer reviewed.
Thus, the study is comprehensive and assess cutting-edge information available in the field of nutritional geometry.
The results of the research are summarized in a single graph
Reply: Thank you for appreciating the conciseness of my short communication paper.
Reviewer 5 Report
Comments and Suggestions for Authors
The author has analyzed protein / carbohydrate intake ratio of larval and adult traits of Drosophila melanogaster, based on the data taken from the Geometric Framework.
The topic of developing models to explore how different protein-to-carbohydrate ratios affect various traits is interesting. But I regret to mention that I cannot support this work. Here are my comments for the author.
- The analysis is more a simple statistical analysis and data visualization on an existing data, resulting in a relatively simple single figure. So, the results are too limited. Also, there is limited depth in the discussion or biological interpretation of this limited results.
- The discussion bolds the potential for nutritional trade-offs; however, it remains speculative. It lacks a deep dive into how these trade-offs specifically manifest at the molecular or genetic level. This drawback limits the contribution of the study to current knowledge.
- This study focuses on existing data from other experiments. It could be fine if novel insights were extracted from that, which is not the case here. So, it may not present groundbreaking findings and not fully contribute new, transformative knowledge.
All in all, the current work is a valuable data analysis study but does not offer much in terms of novelty or deep insights, and I suggest rejecting it, unless:
- The author can add further depth to the interpretation of the results (e.g., deeper biological insights, mechanistic pathways).
- The author can add more achievements in the form of extra figures contributing more substantially to understanding the biology behind the trade-offs.
- The author can revise the manuscript to emphasize more novel findings or a unique contribution to the field.
Author Response
The author has analyzed protein / carbohydrate intake ratio of larval and adult traits of Drosophila melanogaster, based on the data taken from the Geometric Framework.
The topic of developing models to explore how different protein-to-carbohydrate ratios affect various traits is interesting. But I regret to mention that I cannot support this work. Here are my comments for the author.
- The analysis is more a simple statistical analysis and data visualization on an existing data, resulting in a relatively simple single figure. So, the results are too limited. Also, there is limited depth in the discussion or biological interpretation of this limited results.
Reply: I have added a paragraph to highlight the potential mechanisms that can help females navigate these nutritional trade-offs, which addresses the “limited depth in the discussion” highlighted by the reviewer. The section now reads:
“The data collated here highlights a nutritional conundrum for adult flies: how to balance diets that maximize lifespan against those that enhance reproduction. Physiological nutrient requirements vary depending on life stage, environment, genetic background, and stress levels, making nutritional choices inherently multi-factorial [32]. However, Drosophila melanogaster possess a well-characterised molecular pathway that acts as a dietary switch: the sex peptide (SP) pathway. When females mate, they receive a cocktail of seminal fluid proteins from the male’s ejaculate, the most abundant of which is SP. SP binds to its receptor (sex peptide receptor or SPR) in the female reproductive tract and brain, triggering physiological and behavioral changes, including increased protein intake. Thus, mating serves as a clear biological signal that reproduction should be prioritized at least partly through the physiological cascade activated by SP-SPR interaction. Males also express SPR in the brain, but whether it influences their nutritional needs and behavior remains unclear. Drosophila melanogaster males are reproductively active even in the absence of mates, often engaging with inanimate objects, suggesting they may be inherently predisposed to prioritize diets that support reproduction over longevity.
Based on this, I hypothesize that males and females have different baseline nutritional needs: unmated females require a lower PC ratio until mating triggers a shift towards higher PC ratio diets that enhance reproductive output. In contrast, males may be hardwired to consume higher PC ratio diets by default, unless environmental factors (e.g., low female encounter rate) drive a shift towards diets that favor lifespan extension (Figure 2a). This hypothesis has significant implications for sexual conflict over nutrition. In a D. melanogaster population that adheres to Bateman’s principles—which appears ubiquitous—most females are mated, with fitness constrained only by egg production, whereas a smaller proportion of males successfully mate, leaving many unmated. Under these conditions, sexual conflict over nutrition is likely weak because the optimal PC ratio for both sexes tend to converge: mated females shift to higher PC ratios to maximize reproduction, while males already prioritize these diets for reproductive success (Figure 2b). Further studies are needed to test this hypothesis and, more broadly, investigate how population structure, mating systems, and evolutionary pressures, including Bateman’s principles, shape nutritional responses and adaptation.”
- The discussion bolds the potential for nutritional trade-offs; however, it remains speculative. It lacks a deep dive into how these trade-offs specifically manifest at the molecular or genetic level. This drawback limits the contribution of the study to current knowledge.
Reply: Lines 178 to 180 discusses exactly this and even raises this very same question when I stated: “TOR and AMPK pathways are two obvious higher level regulatory pathways to control nutritional trade-offs but what are the genes that modulate nutritional responses and trade-offs downstream of these major pathways [39–42]?” Lines 180 to 199 then discusses what has been known so far.
As mentioned above, I now revised the discussion to incorporate a new paragraph discussing how organisms can navigate these nutritional trade-offs, which addresses the need for more molecular insights.
- This study focuses on existing data from other experiments. It could be fine if novel insights were extracted from that, which is not the case here. So, it may not present groundbreaking findings and not fully contribute new, transformative knowledge.
Reply: I disagree, but thank you for the valuable comments.
All in all, the current work is a valuable data analysis study but does not offer much in terms of novelty or deep insights, and I suggest rejecting it, unless:
- The author can add further depth to the interpretation of the results (e.g., deeper biological insights, mechanistic pathways).
Reply: See response above.
- The author can add more achievements in the form of extra figures contributing more substantially to understanding the biology behind the trade-offs.
Reply: I will not add more figures because there is no need of more figures to convey the main message of the paper.
- The author can revise the manuscript to emphasize more novel findings or a unique contribution to the field.
Reply: See the new paragraph added which propose a new hypothesis to be tested in the field, pushing the field forward in new directions.
Reviewer 6 Report
Comments and Suggestions for Authors
The manuscript entitled 'Nutritional trade-offs in Drosophila melanogaster' provides a comprehensive analysis of nutritional trade-offs in Drosophila melanogaster using data from the Geometric Framework of Nutrition. It addresses an important topic in nutritional ecology and evolutionary biology, providing valuable insights into how different protein/carbohydrate ratios affect different life-history traits at different life stages. The study combines data from several geometric framework studies to identify nutritional trade-offs across a wide range of traits in Drosophila melanogaster. This approach provides a broader understanding than previous studies, which have typically focused on a limited number of traits.
However, before we can consider accepting your work, there are several issues that need to be addressed:
(1) The abstract and introduction effectively set the context for the study.
(2) The inclusion of studies with different diets (liquid vs. solid) and genetic backgrounds is noted. However, the potential confounding effects of these variables should be discussed more thoroughly in the interpretation of the results.
(3) ‘previous analytical methods to reconstruct and analyse GF landscapes [27,28].’ This would benefit from some specific instructions in the M&M.
(4) The results section clearly describes the distribution of peaks and valleys in the nutrient space and their implications for different traits. However, more specific details on the statistical significance of these distributions and the strength of the relationships between PC ratios and trait expression would strengthen the section.
(5) The discussion adequately interprets the results in the context of the existing literature. However, it could benefit from a more detailed exploration of the molecular mechanisms underlying the observed nutritional trade-offs. The author acknowledges this limitation, but could provide a more detailed discussion of how genetic variability might influence the results.
(6) The conclusion succinctly summarizes the key findings and suggests directions for future research. It appropriately emphasizes the need for mechanistic insights and broader taxonomic understanding.
Author Response
The manuscript entitled 'Nutritional trade-offs in Drosophila melanogaster' provides a comprehensive analysis of nutritional trade-offs in Drosophila melanogaster using data from the Geometric Framework of Nutrition. It addresses an important topic in nutritional ecology and evolutionary biology, providing valuable insights into how different protein/carbohydrate ratios affect different life-history traits at different life stages. The study combines data from several geometric framework studies to identify nutritional trade-offs across a wide range of traits in Drosophila melanogaster. This approach provides a broader understanding than previous studies, which have typically focused on a limited number of traits.
However, before we can consider accepting your work, there are several issues that need to be addressed:
- The abstract and introduction effectively set the context for the study.
Reply: Thank you
- The inclusion of studies with different diets (liquid vs. solid) and genetic backgrounds is noted. However, the potential confounding effects of these variables should be discussed more thoroughly in the interpretation of the results.
Reply: This was now done in the discussion section, which reads:
“Our data reflected studies that used a wide range of methodological details, such as genetic lines and food types (solid vs. liquid), which influence nutritional responses. For example, measuring diet intake and compensatory responses when animals are given a solid diet remains challenging. The CAFÉ assay enables precise quantification of dietary intake; however, it provides a highly unnatural way of feeding for the flies, which could lead to other physiological and behavioral changes that affect their response to diets. We still lack a systematic comparison of the nutritional responses of flies with similar genetic backgrounds on different diet types or different genetic backgrounds on the same diet type to properly quantify the impact of the experimental approach on nutritional responses. With a growing number of studies in the literature, it will become possible to quantify these effects and account for their consequences, leading to better interpretation of results. Nevertheless, it is worth noting that GF landscapes for lifespan and reproduction are qualitatively similar among studies using different genetic strains (Figure S1), which partly assuages concerns about the confounding effects of unmeasured factors on the nutritional patterns presented here.”
- ‘previous analytical methods to reconstruct and analyse GF landscapes [27,28].’ This would benefit from some specific instructions in the M&M.
Reply: Done, which now reads:
“Briefly, each GF landscape is segmented from an image obtained in the original publication and then reconstructed with a semi-automated algorithm capable of building a surrogate GF landscape with peaks and valleys in the same region as the original landscape. This allows us to analyse GF landscapes for which the raw data is unavailable (see Morimoto 2024)
- The results section clearly describes the distribution of peaks and valleys in the nutrient space and their implications for different traits. However, more specific details on the statistical significance of these distributions and the strength of the relationships between PC ratios and trait expression would strengthen the section.
Reply: Thank you for this suggestion. Given the caveats of the data (e.g., different strains, diet types) highlighted in the paper and by the reviewer in comment (2) above, I kindly insist that no statistical inference is conducted in this dataset, as there is not enough data to control for known confounding effects.
- The discussion adequately interprets the results in the context of the existing literature. However, it could benefit from a more detailed exploration of the molecular mechanisms underlying the observed nutritional trade-offs. The author acknowledges this limitation but could provide a more detailed discussion of how genetic variability might influence the results.
Reply: I have anow added a paragraph discussing how organisms might navigate these nutritional trade-offs which addresses the need for more discussion on potential molecular mechanisms.
- The conclusion succinctly summarizes the key findings and suggests directions for future research. It appropriately emphasizes the need for mechanistic insights and broader taxonomic understanding.
Reply: Thank you.
Round 2
Reviewer 2 Report
Comments and Suggestions for Authors
The revised manuscript improves on the earlier version. However, the conclusion section (Lines 216-221) needs improvement. I suggest avoiding references in this section. The authors should write about the key findings of the present work. Then, add 1-3 sentences concerning the importance/significance of the work.
Author Response
The revised manuscript improves on the earlier version. However, the conclusion section (Lines 216-221) needs improvement. I suggest avoiding references in this section. The authors should write about the key findings of the present work. Then, add 1-3 sentences concerning the importance/significance of the work.
Reply: Thank you. I have now edited the conclusion section following the reviewer’s suggestions, which reads:
“Through data collection from a range of studies, I demonstrated that larval and adult traits in D. melanogaster have distinct—often conflicting—nutritional requirements, creating the potential for nutritional trade-offs and sexual conflict over nutrition. Specifically, PC ratios that maximized lifespan almost invariably minimized reproductive traits, particularly in females. Future studies should focus on uncovering the mechanisms that trigger and enable dietary switches that optimize different life-history strategies. Moreover, a broader taxonomic perspective is needed to understand how dietary responses are realized and whether phylogenetic patterns across taxa can provide deeper insights into the evolutionary dynamics of nutritional adaptation.”
Reviewer 4 Report
Comments and Suggestions for Authors
The authors justified the use of only eight articles; however, this proves the fragility of the research results based on little information. This is clear in the results section, which summarizes little information in just one graph.
Using secondary data from only eight articles makes it impossible to find patterns that are closer to reality. The researchers' initial idea was to cover the entire genus, but they focused on a single species. Drosophila melanogaster has a wide geographic distribution, and it is difficult to establish absolute patterns and consequently make generalizations with so few studies.
Author Response
The authors justified the use of only eight articles; however, this proves the fragility of the research results based on little information. This is clear in the results section, which summarizes little information in just one graph.
Reply: The reviewer dismisses the amount of data presented in Figure 1, but without full understanding of what is behind the work.
Each point in figure 1 was retrieved from at least one (but often more) GF landscapes. These landscapes had either to be redone from raw data, or reconstructed from published literature, a process which is not trivial and was not available until January 2024 when I developed a method to do so (Morimoto, 2024 Aging Cell). To identify each point as peak or valley, a method had to be developed and applied to each GF landscape twice. This method was also unavailable until I developed it, starting in 2019 (Morimoto and Lihoreau, 2019 Am Nat) and more recently, with the nutrigonometry models (Morimoto et al., 2023 Am Nat).
Thus, Figure 1 is simple and concise thanks to the ability to condense information. I have now added a second figure in the Discussion to highlight the mechanistic insights that can be at play between males and females due to the sex peptide (SP) pathway. Figure 2 now looks like this:

Figure 2. Nutritional switches might assuage the potential for nutritional trade-offs. (a) Upon mating, females shift their dietary preference and consume higher PC ratio diets to support egg production, as a result of the sex peptide (SP) pathway. Males eat higher PC ratio regardless of their mating status, maximizing reproductive traits as opposed to lifespan. I hypothesize that this is the default nutritional behavior in females and males, respectively, and males only shift to PC ratios that maximize lifespan due to environmental conditions, which could lead to males surviving longer achieving higher fitness. An example could be a population where female encounter rates by males is relatively low, and males could benefit from living longer to maximize chances of reproduction. (b) Example of two population factors that can modulate the potential for sexual conflict over nutrition. As the proportion of mated female increase, the absolute different of the PC ratio between males and females decrease (i.e., both eating higher PC ratio diets), in turn leading to lower potential for sexual conflict. Stress, such as for example low encounter rates, could lead males to shift their dietary intake to lower PC ratio diets to live longer and increase chances of findings a mate, increasing and the potential for sexual conflict. This hypothesis remains untested.
Using secondary data from only eight articles makes it impossible to find patterns that are closer to reality. The researchers' initial idea was to cover the entire genus, but they focused on a single species. Drosophila melanogaster has a wide geographic distribution, and it is difficult to establish absolute patterns and consequently make generalizations with so few studies.
Reply: This paper was never about covering an entire genus, and this was made clear from the onset. It is rather annoying that the reviewer thinks that he/she knows more about the motivation of the paper than the author. As mentioned in the response to Round 1, it is impossible to conduct this study in any other species of Drosophila simply because there are not enough studies across different life history traits.
In my previous reply, I also show how consistent the patterns found here are across studies for two major traits (lifespan and reproduction). I have now added this figure as Supplementary material (Figure S1). This means that the patters I describe here are likely applicable to many Drosophila melanogaster lines. We also know that findings in Drosophila melanogaster are similar to findings in other species of flies (e.g., black soldier flies; Barragan-Fonseca et al. 2019; Fanson et al., 2009) and more generally, other insects (e.g., Bunning et al., 2016; Rapkin et al., 2018). Thus, the qualitative results presented in this study can be generalized, although more studies are needed across species (as argued in the discussion and conclusion sections of this manuscript).